# Deep Mean-Shift Priors for Image Restoration

**Siavash A. Bigdeli**
University of Bern
bigdeli@inf.unibe.ch

**Meiguang Jin**
University of Bern
jin@inf.unibe.ch

**Paolo Favaro**
University of Bern
favaro@inf.unibe.ch

**Matthias Zwicker**
University of Bern, and University of Maryland, College Park
zwicker@cs.umd.edu

## Abstract

In this paper we introduce a natural image prior that directly represents a Gaussian-smoothed version of the natural image distribution. We include our prior in a formulation of image restoration as a Bayes estimator that also allows us to solve noise-blind image restoration problems. We show that the gradient of our prior corresponds to the mean-shift vector on the natural image distribution. In addition, we learn the mean-shift vector field using denoising autoencoders, and use it in a gradient descent approach to perform Bayes risk minimization. We demonstrate competitive results for noise-blind deblurring, super-resolution, and demosaicing.

## 1 Introduction

Image restoration tasks, such as deblurring and denoising, are ill-posed problems, whose solution requires effective image priors. In the last decades, several natural image priors have been proposed, including total variation [29], gradient sparsity priors [12], models based on image patches [5], and Gaussian mixtures of local filters [25], just to name a few of the most successful ideas. See Figure 1 for a visual comparison of some popular priors. More recently, deep learning techniques have been used to construct generic image priors.

Here, we propose an image prior that is directly based on an estimate of the natural image probability distribution. Although this seems like the most intuitive and straightforward idea to formulate a prior, only few previous techniques have taken this route [20]. Instead, most priors are built on intuition or statistics of natural images (e.g., sparse gradients). Most previous deep learning priors are derived in the context of specific algorithms to solve the restoration problem, but it is not clear how these priors relate to the probability distribution of natural images. In contrast, our prior directly represents the natural image distribution smoothed with a Gaussian kernel, an approximation similar to using a Gaussian kernel density estimate. Note that we cannot hope to use the true image probability distribution itself as our prior, since we only have a finite set of samples from this distribution. We show a visual comparison in Figure 1, where our prior is able to capture the structure of the underlying image, but others tend to simplify the texture to straight lines and sharp edges.

We formulate image restoration as a Bayes estimator, and define a utility function that includes the smoothed natural image distribution. We approximate the estimator with a bound, and show that the gradient of the bound includes the gradient of the logarithm of our prior, that is, the Gaussian smoothed density. In addition, the gradient of the logarithm of the smoothed density is proportional to the mean-shift vector [8], and it has recently been shown that denoising autoencoders (DAEs) learn such a mean-shift vector field for a given set of data samples [1, 4]. Hence we call our prior a *deep mean-shift prior*, and our framework is an example of Bayesian inference using deep learning.

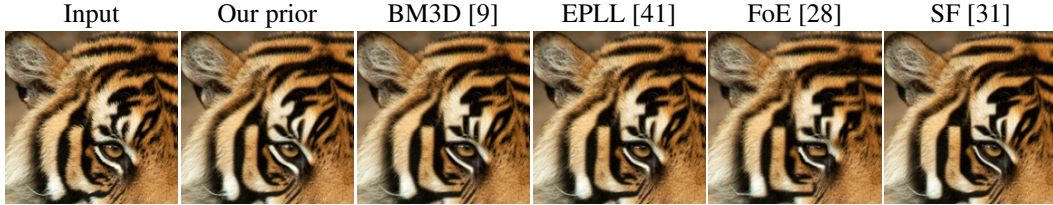

| Input | Our prior | BM3D [9] | EPLL [41] | FoE [28] | SF [31] |

Figure 1: Visualization of image priors using the method by Shaham et al. [32]: Our deep mean-shift prior learns complex structures with different curvatures. Other priors prefer simpler structures like lines with small curvature or sharp corners.

We demonstrate image restoration using our prior for noise-blind deblurring, super-resolution, and image demosaicing, where we solve Bayes estimation using a gradient descent approach. We achieve performance that is competitive with the state of the art for these applications. In summary, the main contributions of this paper are:

- A formulation of image restoration as a Bayes estimator that leverages the Gaussian smoothed density of natural images as its prior. In addition, the formulation allows us to solve noise-blind restoration problems.

- An implementation of the prior, which we call *deep mean-shift prior*, that builds on denoising autoencoders (DAEs). We rely on the observation that DAEs learn a mean-shift vector field, which is proportional to the gradient of the logarithm of the prior.

- Image restoration techniques based on gradient-descent risk minimization with competitive results for noise-blind image deblurring, super-resolution, and demosaicing. [1]

## 2  Related Work

**Image Priors.**   A comprehensive review of previous image priors is outside the scope of this paper. Instead, we refer to the overview by Shaham et al. [32], where they propose a visualization technique to compare priors. Our approach is most related to techniques that leverage CNNs to learn image priors. These techniques build on the observation by Venkatakrishnan et al. [33] that many algorithms that solve image restoration via MAP estimation only need the proximal operator of the regularization term, which can be interpreted as a MAP denoiser [22]. Venkatakrishnan et al. [33] build on the ADMM algorithm and propose to replace the proximal operator of the regularizer with a denoiser such as BM3D [9] or NLM [5]. Unsurprisingly, this inspired several researchers to learn the proximal operator using CNNs [6, 40, 35, 22]. Meinhardt et al. [22] consider various proximal algorithms including the proximal gradient method, ADMM, and the primal-dual hybrid gradient method, where in each case the proximal operator for the regularizer can be replaced by a neural network. They show that no single method will produce systematically better results than the others.

In the proximal techniques the relation between the proximal operator of the regularizer and the natural image probability distribution remains unclear. In contrast, we explicitly use the Gaussian-smoothed natural image distribution as a prior, and we show that we can learn the gradient of its logarithm using a denoising autoencoder.

Romano et al. [27] designed a prior model that is also implemented by a denoiser, but that does not build on a proximal formulation such as ADMM. Interestingly, the gradient of their regularization term boils down to the residual of the denoiser, that is, the difference between its input and output, which is the same as in our approach. However, their framework does not establish the connection between the prior and the natural image probability distribution, as we do. Finally, Bigdeli and Zwicker [4] formulate an energy function, where they use a Denoising Autoencoder (DAE) network for the prior, as in our approach, but they do not address the case of noise-blind restoration.

**Noise- and Kernel-Blind Deconvolution.**   Kernel-blind deconvolution has seen the most effort recently, while we support the fully (noise and kernel) blind setting. Noise-blind deblurring is usually

performed by first estimating the noise level and then restoration with the estimated noise. Jin et al. [14] proposed a Bayes risk formulation that can perform deblurring by adaptively changing the regularization without the need of the noise variance estimate. Zhang et al. [37, 38] explored a spatially-adaptive sparse prior and scale-space formulation to handle noise- or kernel-blind deconvolution. These methods, however, are tailored specifically to image deconvolution. Also, they only handle the noise- or kernel-blind case, but not fully blind.

# 3 Bayesian Formulation

We assume a standard model for image degradation,

$$y = k * \xi + n, \quad n \sim \mathcal{N}(0, \sigma_n^2), \tag{1}$$

where $\xi$ is the unknown image, $k$ is the blur kernel, $n$ is zero-mean Gaussian noise with variance $\sigma_n^2$, and $y$ is the observed degraded image. We restore an estimate $x$ of the unknown image by defining and maximizing an objective consisting of a data term and an image likelihood,

$$\underset{x}{\operatorname{argmax}} \, \Phi(x) = \mathrm{data}(x) + \mathrm{prior}(x). \tag{2}$$

Our core contribution is to construct a prior that corresponds to the logarithm of the Gaussian-smoothed probability distribution of natural images. We will optimize the objective using gradient descent, and leverage the fact that we can learn the gradient of the prior using a denoising autoencoder (DAE). We next describe how we define our objective by formulating a Bayes estimator in Section 3.1, then explain how we leverage DAEs to obtain the gradient of our prior in Section 3.2, describe our gradient descent approach in Section 3.3, and finally our image restoration applications in Section 4.

## 3.1 Defining the Objective via a Bayes Estimator

A typical approach to solve the restoration problem is via a maximum a posteriori (MAP) estimate, where one considers the posterior distribution of the restored image $p(x|y) \propto p(y|x)p(x)$, derives an objective consisting of a sum of data and prior terms by taking the logarithm of the posterior, and maximizes it (minimizes the negative log-posterior, respectively). Instead, we will compute a Bayes estimator $x$ for the restoration problem by maximizing the posterior expectation of a utility function,

$$E_{\tilde{x}}[G(\tilde{x}, x)] = \int G(\tilde{x}, x) p(y|\tilde{x}) p(\tilde{x}) d\tilde{x} \tag{3}$$

where $G$ denotes the utility function (e.g., a Gaussian), which encourages its two arguments to be similar. This is a generalization of MAP, where the utility is a Dirac impulse.

Ideally, we would like to use the true data distribution as the prior $p(\tilde{x})$. But we only have data samples, hence we cannot learn this exactly. Therefore, we introduce a smoothed data distribution

$$p'(x) = E_\eta[p(x + \eta)] = \int g_\sigma(\eta) p(x + \eta) d\eta, \tag{4}$$

where $\eta$ has a Gaussian distribution with zero-mean and variance $\sigma^2$, which is represented by the smoothing kernel $g_\sigma$. The key idea here is that it is possible to estimate the smoothed distribution $p'(x)$ or its gradient from sample data. In particular, we will need the gradient of its logarithm, which we will learn using denoising autoencoders (DAEs). We now define our utility function as

$$G(\tilde{x}, x) = g_\sigma(\tilde{x} - x) \frac{p'(x)}{p(\tilde{x})}. \tag{5}$$

where we use the same Gaussian function $g_\sigma$ with standard deviation $\sigma$ as introduced for the smoothed distribution $p'$. This penalizes the estimate $x$ if the latent parameter $\tilde{x}$ is far from it. In addition, the term $p'(x)/p(\tilde{x})$ penalizes the estimate if its smoothed density is lower than the true density of the latent parameter. Unlike the utility in Jin et al. [14], this approach will allow us to express the prior directly using the smoothed distribution $p'$.

By inserting our utility function into the posterior expected utility in Equation (3) we obtain

$$E_{\tilde{x}}[G(\tilde{x}, x)] = \int g_\sigma(\epsilon) p(y|x + \epsilon) \int g_\sigma(\eta) p(x + \eta) d\eta d\epsilon, \tag{6}$$

where the true density $p(\tilde{x})$ canceled out, as desired, and we introduced the substitution $\epsilon = \tilde{x} - x$.

We finally formulate our objective by taking the logarithm of the expected utility in Equation (6), and introducing a lower bound that will allow us to split Equation (6) into a data term and an image likelihood. By exploiting the concavity of the $\log$ function, we apply Jensen's inequality and get our objective $\Phi(x)$ as

$$
\begin{aligned}
\log E_{\tilde{x}}[G(\tilde{x}, x)] = \log \int g_\sigma(\epsilon) p(y|x + \epsilon) & \int g_\sigma(\eta) p(x + \eta) d\eta d\epsilon \\
\geq \int g_\sigma(\epsilon) \log & \left[ p(y|x + \epsilon) \int g_\sigma(\eta) p(x + \eta) d\eta \right] d\epsilon \\
= \underbrace{\int g_\sigma(\epsilon) \log p(y|x + \epsilon) d\epsilon}_{\text{Data term } \mathrm{data}(x)} + \underbrace{\log \int g_\sigma(\eta) p(x + \eta) d\eta}_{\text{Image likelihood } \mathrm{prior}(x)} = \Phi(x). \quad (7)
\end{aligned}
$$

**Image Likelihood.** We denote the image likelihood as

$$
\mathrm{prior}(x) = \log \int g_\sigma(\eta) p(x + \eta) d\eta. \quad (8)
$$

The key observation here is that our prior expresses the image likelihood as the logarithm of the Gaussian-smoothed true natural image distribution $p(x)$, which is similar to a kernel density estimate.

**Data Term.** Given that the degradation noise is Gaussian, we see that [14]

$$
\mathrm{data}(x) = \int g_\sigma(\epsilon) \log p(y|x + \epsilon) d\epsilon = -\frac{|y - k * x|^2}{2\sigma_n^2} - M \frac{\sigma^2}{2\sigma_n^2} |k|^2 - N \log \sigma_n + \mathrm{const}, \quad (9)
$$

where $M$ and $N$ denote the number of pixels in $x$ and $y$ respectively. This will allow us to address noise-blind problems as we will describe in detail in Section 4.

## 3.2 Gradient of the Prior via Denoising Autoencoders (DAE)

A key insight of our approach is that we can effectively learn the gradients of our prior in Equation (8) using denoising autoencoders (DAEs). A DAE $r_\sigma$ is trained to minimize [34]

$$
\mathcal{L}_{\mathrm{DAE}} = \mathbb{E}_{\eta, x} \left[ |x - r_\sigma(x + \eta)|^2 \right], \quad (10)
$$

where the expectation is over all images $x$ and Gaussian noise $\eta$ with variance $\sigma^2$, and $r_\sigma$ indicates that the DAE was trained with noise variance $\sigma^2$. Note that this is the same loss as in non-parametric least squares estimators [23, 26, 20]. Similar to Alain and Bengio [1], we parametrize this estimator using neural networks for fast evaluation. They show that the output $r_\sigma(x)$ of the optimal DAE (by assuming unlimited capacity) is related to the true data distribution $p(x)$ as

$$
r_\sigma(x) = x - \frac{\mathbb{E}_\eta [p(x - \eta)\eta]}{\mathbb{E}_\eta [p(x - \eta)]} = x - \frac{\int g_\sigma(\eta) p(x - \eta)\eta d\eta}{\int g_\sigma(\eta) p(x - \eta) d\eta} \quad (11)
$$

where the noise has a Gaussian distribution $g_\sigma$ with standard deviation $\sigma$. This is simply a continuous formulation of mean-shift, and $g_\sigma$ corresponds to the smoothing kernel in our prior, Equation (8).

To obtain the relation between the DAE and the desired gradient of our prior, we first rewrite the numerator in Equation (11) using the Gaussian derivative definition to remove $\eta$, that is

$$
\int g_\sigma(\eta) p(x - \eta)\eta d\eta = -\sigma^2 \int \nabla g_\sigma(\eta) p(x - \eta) d\eta = -\sigma^2 \nabla \int g_\sigma(\eta) p(x - \eta) d\eta, \quad (12)
$$

where we used the Leibniz rule to interchange the $\nabla$ operator with the integral. Plugging this back into Equation (11), we have

$$
r_\sigma(x) = x + \frac{\sigma^2 \nabla \int g_\sigma(\eta) p(x - \eta) d\eta}{\int g_\sigma(\eta) p(x - \eta) d\eta} = x + \sigma^2 \nabla \log \int g_\sigma(\eta) p(x - \eta) d\eta. \quad (13)
$$

One can now see that the DAE error, that is, the difference $r_\sigma(x) - x$ between the output of the DAE and its input, is the gradient of the image likelihood in Equation (8). Hence, a main result of our approach is that we can write the gradient of our prior using the DAE error,

$$
\nabla \mathrm{prior}(x) = \nabla \log \int g_\sigma(\eta) p(x + \eta) d\eta = \frac{1}{\sigma^2} \left( r_\sigma(x) - x \right). \quad (14)
$$

| | | | |
|---|---|---|---|
| NB: | **1.** $u^t = \frac{1}{\sigma_n^2} K^T(Kx^{t-1} - y) - \nabla \text{prior}_L^s(x^{t-1})$ | **2.** $\bar{u} = \mu\bar{u} - \alpha u^t$ | **3.** $x^t = x^{t-1} + \bar{u}$ |
| NA: | **1.** $u^t = \lambda^t K^T(Kx^{t-1} - y) - \nabla \text{prior}_L^s(x^{t-1})$ | **2.** $\bar{u} = \mu\bar{u} - \alpha u^t$ | **3.** $x^t = x^{t-1} + \bar{u}$ |
| KE: | **4.** $v^t = \lambda^t \left[ x^T(K^{t-1}x^{t-1} - y) + M\sigma^2 k^{t-1} \right]$ | **5.** $\bar{v} = \mu_k\bar{v} - \alpha_k v^t$ | **6.** $k^t = k^{t-1} + \bar{v}$ |

Table 1: Gradient descent steps for non-blind (NB), noise-blind (NA), and kernel-blind (KE) image deblurring. Kernel-blind deblurring involves the steps for (NA) and (KE) to update image and kernel.

### 3.3 Stochastic Gradient Descent

We consider the optimization as minimization of the negative of our objective $\Phi(x)$ and refer to it as gradient descent. Similar to Bigdeli and Zwicker [4], we observed that the trained DAE is overfitted to noisy images. Because of the large gap in dimensionality between the embedding space and the natural image manifold, the vast majority of training inputs (noisy images) for the DAE lie at a distance very close to $\sigma$ from the natural image manifold. Hence, the DAE cannot effectively learn mean-shift vectors for locations that are closer than $\sigma$ to the natural image manifold. In other words, our DAE does not produce meaningful results for input images that do not exhibit noise close to the DAE training $\sigma$.

To address this issue, we reformulate our prior to perform stochastic gradient descent steps that include noise sampling. We rewrite our prior from Equation (8) as

$$\text{prior}(x) = \log \int g_\sigma(\eta)p(x+\eta)d\eta \tag{15}$$

$$= \log \int g_{\sigma_2}(\eta_2) \int g_{\sigma_1}(\eta_1)p(x+\eta_1+\eta_2)d\eta_1 d\eta_2 \tag{16}$$

$$\geq \int g_{\sigma_2}(\eta_2) \log \left[ \int g_{\sigma_1}(\eta_1)p(x+\eta_1+\eta_2)d\eta_1 \right] d\eta_2 = \text{prior}_L(x), \tag{17}$$

where $\sigma_1^2 + \sigma_2^2 = \sigma^2$, we used the fact that two Gaussian convolutions are equivalent to a single convolution with a Gaussian whose variance is the sum of the two, and we applied Jensen's inequality again. This leads to a new lower bound for the prior, which we call $\text{prior}_L(x)$. Note that the bound proposed by Jin et al. [14] corresponds to the special case where $\sigma_1 = 0$ and $\sigma_2 = \sigma$.

We address our DAE overfitting issue by using the new lower bound $\text{prior}_L(x)$ with $\sigma_1 = \sigma_2 = \frac{\sigma}{\sqrt{2}}$. Its gradient is

$$\nabla \text{prior}_L(x) = \frac{2}{\sigma^2} \int g_{\frac{\sigma}{\sqrt{2}}}(\eta_2) \left( r_{\frac{\sigma}{\sqrt{2}}}(x+\eta_2) - (x+\eta_2) \right) d\eta_2. \tag{18}$$

In practice, computing the integral over $\eta_2$ is not possible at runtime. Instead, we approximate the integral with a single noise sample, which leads to the stochastic evaluation of the gradient as

$$\nabla \text{prior}_L^s(x) = \frac{2}{\sigma^2} \left( r_{\frac{\sigma}{\sqrt{2}}}(x+\eta_2) - x \right), \tag{19}$$

where $\eta_2 \sim \mathcal{N}(0, \sigma_2^2)$. This addresses the overfitting issue, since it means we add noise each time before we evaluate the DAE. Given the stochastically sampled gradient of the prior, we apply a gradient descent approach with momentum that consists of the following steps:

$$\boxed{\textbf{1.}\ u^t = -\nabla \text{data}(x^{t-1}) - \nabla \text{prior}_L^s(x^{t-1}) \quad \textbf{2.}\ \bar{u} = \mu\bar{u} - \alpha u^t \quad \textbf{3.}\ x^t = x^{t-1} + \bar{u}} \tag{20}$$

where $u^t$ is the update step for $x$ at iteration $t$, $\bar{u}$ is the running step, and $\mu$ and $\alpha$ are the momentum and step-size.

## 4 Image Restoration using the Deep Mean-Shift Prior

We next describe the detailed gradient descent steps, including the derivatives of the data term, for different image restoration tasks. We provide a summary in Table 1. For brevity, we omit the role of downsampling (required for super-resolution) and masking.

|  | | Levin [19] | | | | Berkeley [2] | | | |
|---|---|---|---|---|---|---|---|---|---|
| Method | $\sigma_n$: | 2.55 | 5.10 | 7.65 | 10.2 | 2.55 | 5.10 | 7.65 | 10.2 |
| FD [18] | | 30.03 | 28.40 | 27.32 | 26.52 | 24.44 | 23.24 | 22.64 | 22.07 |
| EPLL [41] | | 32.03 | 29.79 | 28.31 | 27.20 | 25.38 | 23.53 | 22.54 | 21.91 |
| RTF-6 [30]* | | 32.36 | 26.34 | 21.43 | 17.33 | 25.70 | 23.45 | 19.83 | 16.94 |
| CSF [31] | | 29.85 | 28.13 | 27.28 | 26.70 | 24.73 | 23.61 | 22.88 | 22.44 |
| DAEP [4] | | **32.64** | 30.07 | 28.30 | 27.15 | 25.42 | 23.67 | 22.78 | 22.21 |
| IRCNN [40] | | 30.86 | 29.85 | 28.83 | 28.05 | 25.60 | 24.24 | 23.42 | 22.91 |
| EPLL [41] + NE | | 31.86 | 29.77 | 28.28 | 27.16 | 25.36 | 23.53 | 22.55 | 21.90 |
| EPLL [41] + NA | | 32.16 | **30.25** | 28.96 | 27.85 | 25.57 | 23.90 | 22.91 | 22.27 |
| TV-L2 + NA | | 31.05 | 29.14 | 28.03 | 27.16 | 24.61 | 23.65 | 22.90 | 22.34 |
| GradNet 7S [14] | | 31.43 | 28.88 | 27.55 | 26.96 | 25.57 | 24.23 | 23.46 | 22.94 |
| Ours | | 29.68 | 29.45 | 28.95 | **28.29** | 25.69 | 24.45 | 23.60 | **22.99** |
| Ours + NA | | 32.57 | 30.21 | **29.00** | 28.23 | **26.00** | **24.47** | **23.61** | 22.97 |

Table 2: Average PSNR ($dB$) for non-blind deconvolution on two datasets (*trained for $\sigma_n = 2.55$).

**Non-Blind Deblurring (NB).** The gradient descent steps for non-blind deblurring with a known kernel and degradation noise variance are given in Table 1, top row (NB). Here $K$ denotes the Toeplitz matrix of the blur kernel $k$.

**Noise-Adaptive Deblurring (NA).** When the degradation noise variance $\sigma_n^2$ is unknown, we can solve Equation (9) for the optimal $\sigma_n^2$ (since it is independent of the prior), which gives

$$\sigma_n^2 = \frac{1}{N} \left[ |y - k * x|^2 + M\sigma^2 |k|^2 \right]. \tag{21}$$

By plugging this back into the equation, we get the following data term

$$\text{data}(x) = -\frac{N}{2} \log \left[ |y - k * x|^2 + M\sigma^2 |k|^2 \right], \tag{22}$$

which is independent of the degradation noise variance $\sigma_n^2$. We show the gradient descent steps in Table 1, second row (NA), where $\lambda^t = N\left(|y - Kx^{t-1}|^2 + M\sigma^2|k|^2\right)^{-1}$ adaptively scales the data term with respect to the prior.

**Noise- and Kernel-Blind Deblurring (NA+KE).** Gradient descent in noise-blind optimization includes an intuitive regularization for the kernel. We can use the objective in Equation (22) to jointly optimize for the unknown image and the unknown kernel. The gradient descent steps to update the image remain as in Table 1, second row (NA), and we take additional steps to update the kernel estimate, as in Table 1, third row (KE). Additionally, we project the kernel by applying $k^t = \max(k^t, 0)$ and $k^t = \frac{k^t}{|k^t|_1}$ after each step.

## 5 Experiments and Results

Our DAE uses the neural network architecture by Zhang et al. [39]. We generated training samples by adding Gaussian noise to images from ImageNet [10]. We experimented with different noise levels and found $\sigma_1 = 11$ to perform well for all our deblurring and super-resolution experiments. Unless mentioned, for image restoration we always take 300 iterations with step length $\alpha = 0.1$ and momentum $\mu = 0.9$. The runtime of our method is linear in the number of pixels, and our implementation takes about 0.2 seconds per iteration for one megapixel on an Nvidia Titan X (Pascal).

### 5.1 Image Deblurring: Non-Blind and Noise-Blind

In this section we evaluate our method for image deblurring using two datasets. Table 2 reports the average PSNR for 32 images from the Levin et al. [19] and 50 images from the Berkeley [2] segmentation dataset, where 10 images are randomly selected and blurred with 5 kernels as in Jin et al. [14]. We highlight the best performing PSNR in bold and underline the second best value. The

| Ground Truth | EPLL [41] | DAEP [4] | GradNet 7S [14] | Ours | Ours + NA |
|---|---|---|---|---|---|

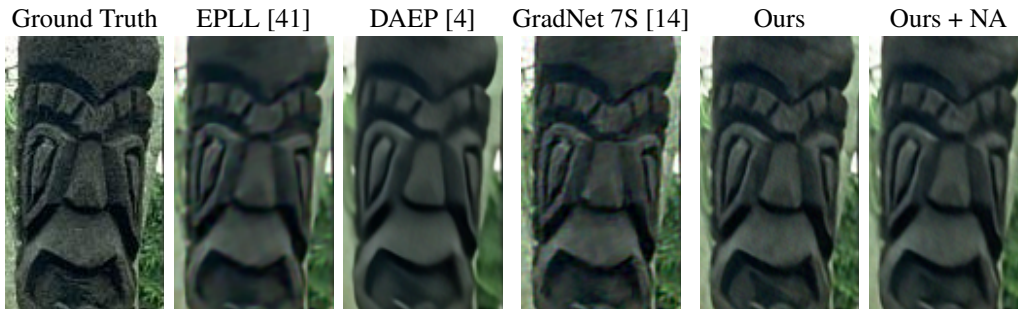

Figure 2: Visual comparison of our deconvolution results.

| Ground Truth | Blurred with 1% noise | Ours (blind) | SSD Error Ratio |
|---|---|---|---|

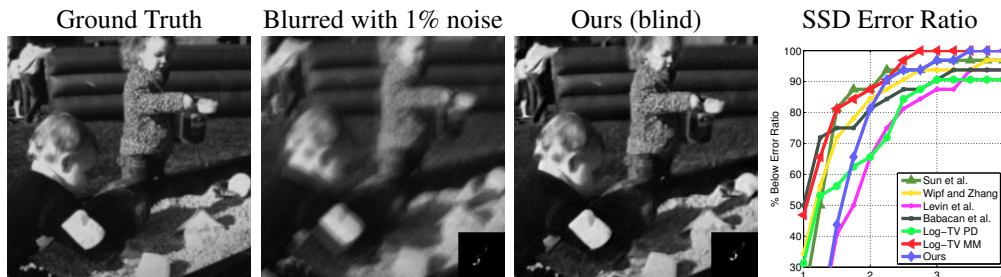

Figure 3: Performance of our method for fully (noise- and kernel-) blind deblurring on Levin's set.

upper half of the table includes non-blind methods for deblurring. EPLL [41] + NE uses a noise estimation step followed by non-blind deblurring. Noise-blind experiments are denoted by NA for noise adaptivity. We include our results for non-blind (Ours) and noise-blind (Ours + NA). Our noise adaptive approach consistently performs well in all experiments and on average we achieve better results than the state of the art. Figure 2 provides a visual comparison of our results. Our prior is able to produce sharp textures while also preserving the natural image structure.

## 5.2 Image Deblurring: Noise- and Kernel-Blind

We performed fully blind deconvolution with our method using Levin et al.'s [19] dataset. In this test, we performed 1000 gradient descent iterations. We used momentum $\mu = 0.7$ and step size $\alpha = 0.3$ for the unknown image and momentum $\mu_k = 0.995$ and step size $\alpha_k = 0.005$ for the unknown kernel. Figure 3 shows visual results of fully blind deblurring and performance comparison to state of the art (last column). We compare the SSD error ratio and the number of images in the dataset that achieves error ratios less than a threshold. Results for other methods are as reported by Perrone and Favaro [24]. Our method can reconstruct all the blurry images in the dataset with errors ratios less than 3.5. Note that our optimization performs end-to-end estimation of the final results and we do not use the common two stage blind deconvolution (kernel estimation, followed by non-blind deconvolution). Additionally our method uses a noise adaptive scheme where we do not assume knowledge of the input noise level.

## 5.3 Super-resolution

To demonstrate the generality of our prior, we perform an additional test with single image super-resolution. We evaluate our method on the two common datasets Set5 [3] and Set14 [36] for different upsampling scales. Since these tests do not include degradation noise ($\sigma_n = 0$), we perform our optimization with a rough weight for the prior and decrease it gradually to zero. We compare our method in Table 3. The upper half of the table represents methods that are specifically trained for super-resolution. SRCNN [11] and TNRD [7] have separate models trained for $\times 2, 3, 4$ scales, and we used the model for $\times 4$ to produce the $\times 5$ results. VDSR [16] and DnCNN-3 [39] have a single model trained for $\times 2, 3, 4$ scales, which we also used to produce $\times 5$ results. The lower half of the table represents general priors that are not designed specifically for super-resolution. Our method performs on par with state of the art methods over all the upsampling scales.

| Method | scale: | Set5 [3] | | | | Set14 [36] | | | |
|---|---|---|---|---|---|---|---|---|---|
| | | ×2 | ×3 | ×4 | ×5 | ×2 | ×3 | ×4 | ×5 |
| Bicubic | | 31.80 | 28.67 | 26.73 | 25.32 | 28.53 | 25.92 | 24.44 | 23.46 |
| SRCNN [11] | | 34.50 | 30.84 | 28.60 | 26.12 | 30.52 | 27.48 | 25.76 | 24.05 |
| TNRD [7] | | 34.62 | 31.08 | 28.83 | 26.88 | 30.53 | 27.60 | 25.92 | 24.61 |
| VDSR [16] | | 34.50 | 31.39 | 29.19 | 25.91 | 30.72 | 27.81 | 26.16 | 24.01 |
| DnCNN-3 [39] | | 35.20 | **31.58** | **29.30** | 26.30 | 30.99 | 27.93 | **26.25** | 24.26 |
| DAEP [4] | | **35.23** | 31.44 | 29.01 | 27.19 | **31.07** | **27.93** | 26.13 | 24.88 |
| IRCNN [40] | | 35.07 | 31.26 | 29.01 | 27.13 | 30.79 | 27.68 | 25.96 | 24.73 |
| Ours | | 35.16 | 31.38 | 29.16 | **27.38** | 30.99 | 27.90 | 26.22 | **25.01** |

Table 3: Average PSNR ($dB$) for super-resolution on two datasets.

| Matlab [21] | RTF [15] | Gharbi et al. [13] | Gharbi et al. [13] f.t. | SEM [17] | Ours |
|---|---|---|---|---|---|
| 33.9 | 37.8 | 38.4 | 38.6 | 38.8 | 38.7 |

Table 4: Average PSNR ($dB$) in linear RGB space for demosaicing on the Panasonic dataset [15].

## 5.4 Demosaicing

We finally performed a demosaicing experiment on the dataset introduced by Khashabi et al. [15]. This dataset is constructed by taking RAW images from a Panasonic camera, where the images are downsampled to construct the ground truth data. Due to the down sampling effect, in this evaluation we train a DAE with $\sigma_1 = 3$ noise standard deviation. The test dataset consists of 100 noisy images captured by a Panasonic camera using a Bayer color filter array (RGGB). We initialize our method with Matlab's demosaic function [21]. To get even better initialization, we perform our initial optimization with a large degradation noise estimate ($\sigma_n = 2.5$) and then perform the optimization with a lower estimate ($\sigma_n = 1$). We summarize the quantitative results in Table 4. Our method is again on par with the state of the art. Additionally, our prior is not trained for a specific color filter array and therefore is not limited to a specific sub-pixel order. Figure 4 shows a qualitative comparison, where our method produces much smoother results compared to the previous state of the art.

| Ground Truth | RTF [15] | Gharbi et al. [13] | SEM [17] | Ours |
|---|---|---|---|---|

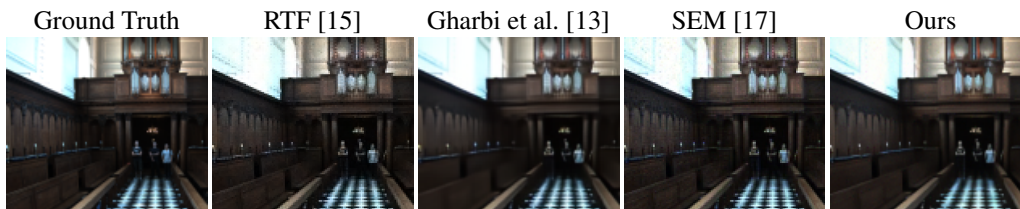

Figure 4: Visual comparison for demosaicing noisy images from the Panasonic data set [15].

## 6 Conclusions

We proposed a Bayesian deep learning framework for image restoration with a generic image prior that directly represents the Gaussian smoothed natural image probability distribution. We showed that we can compute the gradient of our prior efficiently using a trained denoising autoencoder (DAE). Our formulation allows us to learn a single prior and use it for many image restoration tasks, such as noise-blind deblurring, super-resolution, and image demosaicing. Our results indicate that we achieve performance that is competitive with the state of the art for these applications. In the future, we would like to explore generalizing from Gaussian smoothing of the underlying distribution to other types of kernels. We are also considering multi-scale optimization where one would reduce the Bayes utility support gradually to get a tighter bound with respect to maximum a posteriori. Finally, our approach is not limited to image restoration and could be exploited to address other inverse problems.

**Acknowledgments.** MJ and PF acknowledge support from the Swiss National Science Foundation (SNSF) on project 200021-153324.

## Footnotes

[1]The source code of the proposed method is available at `https://github.com/siavashbigdeli/DMSP`.

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
