[Supplementary Material · DMSPrior_supplemental.pdf]

# Deep Mean-Shift Priors for Image Restoration
## -Supplemental Material-

**Siavash A. Bigdeli**
University of Bern
bigdeli@inf.unibe.ch

**Meiguang Jin**
University of Bern
jin@inf.unibe.ch

**Paolo Favaro**
University of Bern
favaro@inf.unibe.ch

**Matthias Zwicker**
University of Bern, and University of Maryland, College Park
zwicker@cs.umd.edu

This supplemental material contains some additional theoretical derivations and experimental results. In Section 1 we show the relationship of our formulation with MAP estimation. We report results on an experiment on runtime additive noise in Section 2. And finally, we provide some additional results for non-blind and noise-blind image deblurring on the dataset by Sun et al. [6] in Section 3.

## 1   Relationship to MAP

Here we show how our estimator relates to MAP and the formulation by Jin et al. [3]. We start with the logarithm of the maximum a-posteriori (MAP) estimator and see that our proposed formulation is bounded from above by MAP. In addition, we observe that our formulation is bounded from below by Jin et al. [3],

$$\log \max_{\hat{x}} p(y|\hat{x})p(\hat{x}) \qquad\qquad\qquad\qquad \text{(MAP)}$$

$$= \max_{x} \log \max_{\hat{x}} p(y|\hat{x})p(\hat{x}) \int g_\sigma(x - \bar{x}_1)d\bar{x}_1 \int g_\sigma(x - \bar{x}_2)d\bar{x}_2$$

$$\geq \max_{x} \log \int g_\sigma(x - \bar{x}_1)p(y|\bar{x}_1)d\bar{x}_1 \int g_\sigma(x - \bar{x}_2)p(\bar{x}_2)d\bar{x}_2$$

$$\geq \max_{x} \int g_\sigma(x - \bar{x}_1)\log p(y|\bar{x}_1)d\bar{x}_1 + \log \int g_\sigma(x - \bar{x}_2)p(\bar{x}_2)d\bar{x}_2 \quad \text{(Our lower bound, Eq. 7)}$$

$$\geq \max_{x} \int g_\sigma(x - \bar{x}_1)\log p(y|\bar{x}_1)d\bar{x}_1 + \int g_\sigma(x - \bar{x}_2)\log p(\bar{x}_2)d\bar{x}_2, \quad \text{(Jin et al. [3])}$$

where we applied Jensen's inequality several times. The interesting observation here is that our formulation is produced by separately relaxing the posteriori and prior, which later allows us to get a tighter lower bound for MAP compared to Jin et al. [3].

## 2   Ratio between Runtime Noise and Training Noise

We perform an evaluation to find the best ratio between the runtime additive noise during stochastic gradient descent and the noise used during DAE training. Specifically, we set up an experiment for image deconvolution with our framework for fixed $\sigma = 15$ and compare the performance for different ratios between $\sigma_1$ and $\sigma_2$ (Equation 17 in the paper). First, we train different DAEs with noise levels $\sigma_1 = 1 : 15$. Second, for each DAE we compute the variance of runtime additive noise by setting it to $\sigma_2^2 = \sigma^2 - \sigma_1^2$. And finally, we evaluate the performance of each configuration with our experiment. Figure 1 shows the quantitative performance for each ratio $\frac{\sigma_1^2}{\sigma^2}$. The configuration $\sigma_1^2 = \sigma_2^2 = \frac{\sigma^2}{2}$ achieves the best performance. This is expected since our the DAE trained with a noise variance $\sigma_1^2$ performs better for that specific noise variance, therefore it is better to use the same variance for runtime additive noise.

Figure 1: Performance comparison for different additive noise variances for stochastic gradient descent. We show the average RMSE for deblurring over a set of images for each ratio $\sigma_1^2/\sigma^2$. As expected, the desired variance $\sigma^2$ should be evenly split over the trained DAE and additive noise during stochastic gradient descent, that is $\sigma_1^2/\sigma^2 = \sigma_2^2/\sigma^2 = 0.5$.

| Method | $\sigma \rightarrow$ | 2.55 | 5.10 | 7.65 | 10.2 |
|---|---|---|---|---|---|
| FD [4] | | 30.79 | 28.90 | 27.86 | 27.14 |
| EPLL [11] | | 32.05 | 29.60 | 28.25 | 27.34 |
| CSF [5] | | 30.88 | 28.60 | 27.65 | 26.97 |
| TNRD [2] | | 30.03 | 28.79 | - | - |
| DAEP [1] | | 31.76 | 29.31 | 28.01 | 27.16 |
| IRCNN [9] | | 31.80 | **30.13** | **28.93** | **28.09** |
| EPLL [11] + NE | | 32.02 | 29.60 | 28.25 | 27.34 |
| EPLL [11] + NA | | **32.18** | 30.08 | 28.77 | 27.81 |
| TV-L2 + NA | | 30.07 | 28.59 | 27.60 | 26.89 |
| GradNet 7S [3] | | 31.75 | 29.31 | 28.04 | 27.54 |
| Ours | | 29.41 | 29.04 | 28.56 | 27.97 |
| Ours + NA | | 32.01 | 29.56 | 28.56 | 27.93 |

Table 1: Average PSNR ($dB$) for non-blind deconvolution on the Sun et al. [6] dataset.

## 3 Image Deblurring: Additional Results

We performed additional experiments on 640 images from Sun et al. [6]. We provide the results in Table 1 for non-blind and noise-blind experiments. Similar to other experiments, our noise adaptive approach (Ours + NA) achieves state of the art results in this dataset.

We also compare visual results for real camera noise and motion blur in Figure 2. Our noise- and kernel-blind optimization (NA + KE) is robust to real camera noise and non-uniform motion blur.

| Noise and Blur | Tai et al. [7] | Zhong et al. [10] | Zhang et al. [8] | Ours |
|---|---|---|---|---|

Figure 2: Visual comparison for restoration from real camera noise and blur.