[Reviews · NeurIPS 2017]

Reviewer 1



This is a well-written paper that introduces an approach to image restoration based on a "hypothetical" prior, the gradient of which can be approximated by a trained denoising autoencoder. They key insight behind the paper is that if you construct a natural image prior in a particular way (specifically, if the natural image distribution is smoothed with a Gaussian kernel), then the gradient of this prior corresponds precisely to the error of an optimal denoising autoencoder (that is, the difference between the output of the autoencoder and its input). In this way, the prior in itself never actually needs to be instantiated - instead, one trains an autoencoder at a fixed noise level (corresponding to the assumed Gaussian noise level of the corrupting process), and then, during inference, repeatedly evaluates the autoencoder at each step of the gradient descent procedure for image restoration to obtain the gradient at the current iterate. The authors devise update steps for non-blind, noise-blind, as well as kernel-blind image deblurring. One practical problem is that the autoencoder is trained for a fixed noise level. The authors observe that the autoencoder does not produce meaningful results for input images that do not exhibit noise close to the training variance. The proposed counter measure effectively amounts to adding Gaussian noise at a certain variance to the auto-encoder input (the current gradient descent iterate) at each step. Despite being motivated by a lower bound involving noise sampling, this step is somewhat ad-hoc. Question to the authors: Is this only need if the corrupted input image exposes a noise level other than the one the autoencoder was trained for? Or is the gradient descent procedure itself producing less and less noisy iterates, so that even if the original corrupted image had the "right" noise level, the "unmodified" iterates become unsuitable for the autoencoder eventually, preventing further progress? The authors present applications of their approach to 1) image deblurring (non-blind and noise-blind), 2) image deblurring (noise- and kernel-blind), 3) super-resolution, and 4) demosaicing. The experiments seem very thorough in general and demonstrate appealing results. I am a bit puzzled about Table 2, however - specifically the performance of RTF-6 [28] as a function of the noise variance. In table 2, the PSNR of RTF-6 drops dramatically as the noise level is increased. To some extent this is normal (all methods suffer from higher noise level), but the dramatic drop of RTF-6 (32.36 at \sigma=2.55 to 21.43 at \sigma=7.65) is unexpected and not consistent with [28]. It is worth pointing out that while the RTF cascade is trained for a fixed noise level, there is a parameter \alpha of the inference routine that must be adjusted to match the noise level of the input. It seems to me that this was not done when the results for Table 2 were produced. I would recommend getting in touch with U. Schmidt to ensure his RTF code was properly used. Overall, I find that this paper establishes interesting connections and hence has some theoretical appeal. Moreover, the authors clearly demonstrate that the approach is applicable to a wide range of image restoration tasks and yields high accuracy. Hence, I strongly recommend that the paper be accepted.

Reviewer 2



Summary ------- The paper proposes a way to do Bayesian estimation for image restoration problems by using gradient descent to minimize a bound of the Bayes risk. In particular, the paper uses Gaussian smoothing in the utility function and the image prior, which allows to establish a connection between the gradient of the log-prior and denoising auto-encoders (DAEs), which are used to learn the prior from training data. Several image restoration experiments demonstrate results on par with the state-of-the-art. Main comments ------------- The positive aspects of the paper are its good results (on par with state-of-the-art methods) for several image restoration tasks and the ability of the same learned prior to be used for several tasks (mostly) without re-training. Another strong aspect is the ability to jointly do noise- and kernel-blind deconvolution, which other methods typically cannot do. I found it somewhat difficult to judge the novelty of the paper, since it seems to be heavily influenced/based on the papers by Jin et al. [14] (noise-adaptive approach, general derivation approach) and Bigdeli and Zwicker [4] (using DAEs as priors, mean-shift connection). Please properly attribute where things have been taken from previous work and make the differences to the proposed approach more clear. In particular: - The mean-shift connection is not really discussed, just written that it is "well-known". Please explain this in more detail, or just remove it from the paper (especially from the title). - Using a Gaussian function as/in the utility function seems to be taken from [14] (?), but not acknowledged. The entire (?) noise-adaptive approach seems to be adopted from [14], please acknowledge/distinguish more thoroughly (e.g. in lines 168-171). Although the experiments demonstrate that the proposed method can lead to good results, many approximations and bounds are used to make it work. Can you give an explanation why the method works so well despite this? Related to this, if I understand correctly, the whole noise sampling approach in section 3.3 is only used to make up for the fact that the learned DAE is overfitted to noisy images? I don't understand how this fixes the problem. Regarding generalization, why is it necessary to train a different DAE for the demosaicing experiments? Shouldn't the image prior be generic? Related to this, the noise- and kernel-blind experiments (section 5.2) use different gradient descent parameters. How sensitive is the method towards these hyper-parameters? The paper doesn't mention at all if the method has reasonable runtimes for typical image sizes. Please provide some ballpark test runtimes for a single gradient descent step (for the various restoration tasks). Miscellaneous ------------- - I found the abstract hard to understand without having read the rest of the paper. - It does seems somewhat artificial to include p'(x)/p(\tilde{x}) in Eq. 5 instead of just removing this term from the utility function and simply replacing p(x) with the smoothed prior p'(x). - Typo in line 182: "form" -> "from" - Please define the Gaussian function g_\sigma in Eq. 6 and later. - Table 2: - EPLL/TV-L2 + NA not explained, how can they use the noise-adaptive approach? - Misleading results: e.g. RTF-6 [28] only trained for \sigma=2.55, doesn't make sense to evaluate for other noise levels. Please just leave the table entry blank.

Reviewer 3



This paper is outside my expertise, which is why I did not bid to review it! It proposes to use a denoising autoencoder (DAE) indirectly - not as the main network, but to provide a differentiable regularisation cost which approximates a prior over natural data points. The connection between DAE, mean-shift and the distribution of natural data points apparently comes from prior work, though is used in a novel way here. The paper is clearly written, complete and interesting. Section 5.4 is an exception to "clearly written". Why do the authors take so much time to explain what type of camera was used when a third party created a dataset? What does "Matlab's demosic function" do, and how was it used here for initialisation? Is this task identical to denoising/deblurring but with a different type of noise, or is there something different about it? If the former, then why are the results reported differently (linear RGB PSNR, rather than dB as elsewhere)? It is not clear to this reviewer how much contribution comes directly from the DAE. I presume that a DAE alone will do a poor job at restoring a blurred image, but it may be contributing something towards the evaluation criterion, so I wonder if it would be worth providing the evaluation statistic for the DAE used alone.